# Mixture discrimination training induces durable and generalizable olfactory learning independent of odorant structure and concentration

Xiaoyue Chang[1,2,3], Huibang Tan[1,2], Jiehui Niu[2], Kaiqi Yuan[1,2], Rui Chen[1,2], Wen Zhou[1,2]*

[1]State Key Laboratory of Cognitive Science and Mental Health, Institute of Psychology, Chinese Academy of Sciences, Beijing, China; [2]Department of Psychology, University of Chinese Academy of Sciences, Beijing, China; [3]School of Artificial Intelligence, University of Chinese Academy of Sciences, Beijing, China

**Abstract** Previously, we showed that adult human olfaction retains plasticity in the unilateral processing of molecular chirality (Feng and Zhou, 2019). Using a similar unilateral discrimination protocol across three experiments with human adults (n = 96; 1295 sessions), we now reveal distinct patterns of specificity, generalization, and persistence in olfactory learning, independent of adaptation or task difficulty. Training with binary odor mixtures at varying ratios consistently produced durable gains that transferred across nostrils and generalized to novel mixtures differing in both structure and perceptual quality. Conversely, training with odor enantiomers or concentration differences yielded neither transfer nor generalization, and concentration discrimination learning was short-lived. These results indicate that mixture configural quality is a distinct olfactory attribute from chirality or relative concentration, and that discrimination learning engages plasticity at different stages of olfactory processing depending on the task-relevant attribute. Moreover, they identify mixture discrimination training as a promising strategy for rehabilitating smell loss and cultivating olfactory expertise.

*For correspondence:
zhouwen@ucas.ac.cn

## Editor's evaluation

This important and well controlled study explores the specificity of olfactory perceptual learning. In keeping with previous work, the authors found that learning to discriminate between two enantiomers does not generalize across the nostrils or to unrelated enantiomers, whereas learning to discriminate odor mixtures does generalize across the nostrils and to other odor mixtures, with this learning effect persisting over at least two weeks. The evidence presented to support these findings is convincing, and they will be of interest to scientists working on olfactory perception and learning.

## Introduction

Olfactory training is recommended for patients with olfactory loss resulting from infectious, traumatic, or neurodegenerative conditions (*Hummel et al., 2017*). Interest in this intervention has grown substantially following the surge of olfactory dysfunction associated with the COVID-19 pandemic (*Tan et al., 2022*). The standard protocol in both clinical research and practice typically involves twice-daily exposure to four single compounds—phenylethyl alcohol, eucalyptol, citronellal, and eugenol—with the odors of rose, eucalyptus, lemon, and clove, respectively, for a period of 12 weeks or longer

(*Hummel et al., 2009*). However, the efficacy of this regimen is not consistently evident (*Lechner et al., 2022*) and is sometimes attributed to spontaneous recovery or placebo effect (*Fornazieri et al., 2020*; *Yan, 2023*). The mechanisms underlying improvement remain poorly understood, though it is generally assumed that training promotes neural plasticity, particularly adult neurogenesis within the peripheral olfactory system (e.g., olfactory epithelium) (*Hummel et al., 2017*; *Moreno et al., 2009*; *Wang et al., 2004*).

By contrast, the phenomenology and mechanisms of visual perceptual learning have been extensively studied (*Fahle, 2005*; *Gilbert and Li, 2012*; *Lu and Dosher, 2022*; *Seitz and Dinse, 2007*; *Watanabe and Sasaki, 2015*), showing that it occurs at multiple levels and persists over weeks and months. Its magnitude, specificity, and transfer depend on factors such as stimulus characteristics, task demands, reinforcement, attention, and training protocol. Patterns of specificity, in turn, provide insight into the locus of neural plasticity. This body of work has illuminated the neural basis of perceptual learning and informed the development of rehabilitation strategies for amblyopia, myopia, and low vision (*Chung, 2011*; *Deveau et al., 2013*; *Durrie and McMinn, 2007*; *Rodán et al., 2022*).

To date, relatively few laboratory studies have investigated the specificity and transfer of olfactory perceptual learning. Repeated exposure of one nostril to the steroid androstenone improved detection accuracy in both nostrils among individuals initially unable to perceive its odor (*Mainland et al., 2002*). Repeated threshold testing with everyday odorants enhanced olfactory sensitivity in an odorant-specific manner, but only among women of reproductive age (*Dalton et al., 2002*). Prolonged discrimination training with feedback, using a pair of odor enantiomers in one nostril, improved chiral discrimination, but the effect did not generalize to the untrained nostril or to structurally dissimilar odor enantiomers (in terms of the chiral center) (*Feng and Zhou, 2019*). Notably, these studies did not assess the durability of learning effects, and no evidence has been reported for generalization to odorants unrelated to the training material.

To maximize the clinical utility of olfactory training, it preferably should generalize to untrained odorants and confer long-term benefits. This requires learning to occur at a representation level that is invariant to nostril of origin and to chemical or perceptual differences between trained and untrained odorants—that is, engagement of high-order olfactory processing. Guided by this rationale, we examined the specificity, transfer, and persistence of olfactory perceptual learning using odor mixture discrimination. We selected discrimination tasks over passive exposure and odor mixtures over single compounds because complex tasks and stimuli are more likely to recruit high-level olfactory mechanisms and thereby promote transfer. Moreover, most real-world odors are mixtures rather than isolated compounds. For comparison, we also conducted discrimination training with odor enantiomers and with single-compound solutions at varying concentrations and assessed the resulting learning effects.

## Results

### Divergent specificity and transfer, with shared persistence, in mixture and enantiomer discrimination learning

The experimental procedure comprised three phases (*Figure 1A*): baseline, unilateral olfactory training, and post-training testing. In Experiment 1, 24 participants (12 men and 12 women) were randomly assigned to two groups of 12. Over several weeks, one group was trained and tested with odor mixtures (mixture group), and the other with odor enantiomers (enantiomer group).

The olfactory stimuli for the mixture group consisted of two pairs of binary mixtures (*Figure 1B*): a:b and b:a mixtures of guaiacol (1% v/v in propylene glycol) and eugenol (1% v/v), and a:b and b:a mixtures of 2-butanol (1% v/v) and 2-heptanol (1% v/v). For three participants, the ratio a:b was 9:11; for the remaining nine, it was 7:9. The mixture constituents were structurally similar and comparable in olfactory intensity and valence (intensity: $ts_{23}$ = 0.57 and 1.68, ps = 0.58 and 0.11; valence: $ts_{23}$ = –0.84 and 0, ps = 0.41 and 1; ratings from an independent panel of 24 odor judges; *Figure 1—figure supplement 1A*), yet perceptually odor-dissimilar (mean similarity ratings ± SDs = 3.0 ± 1.9 and 2.7 ± 1.5 on a 7-point Likert scale, where 7 = extremely similar). Although guaiacol/eugenol (phenols) are structurally distinct from 2-butanol/2-heptanol (aliphatic alcohols), across all pairwise comparisons among the four compounds, perceptual similarity did not differ between structurally similar and dissimilar pairs (2.9 ± 1.5 vs. 2.9 ± 0.9; $t_{23}$ = –0.035, p = 0.97; *Figure 1—figure supplement 1B*). Five

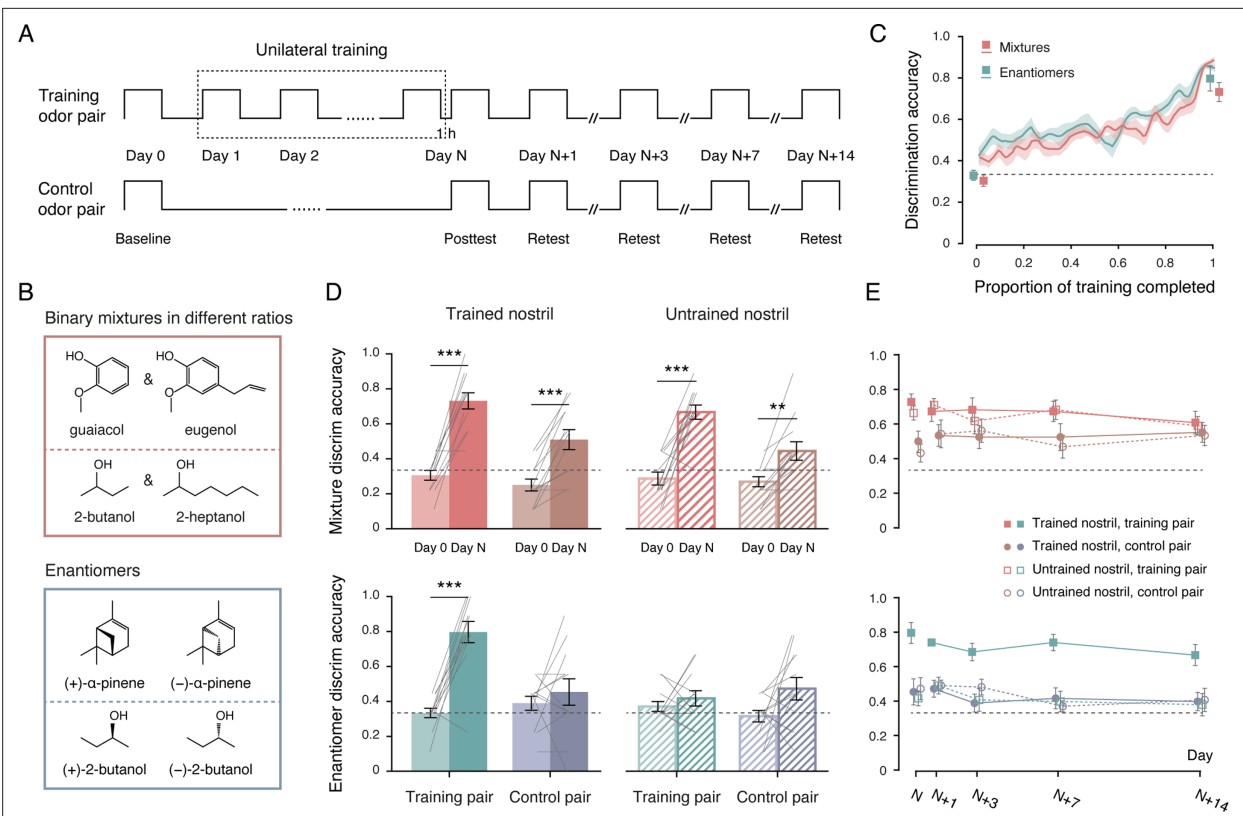

**Figure 1.** Experiment 1: distinct patterns of specificity and transfer in mixture and enantiomer discrimination learning. (**A**) Schematic illustration of the experimental procedure, consisting of three phases: baseline (Day 0), unilateral olfactory training (Day 1 to Day N), and post-training testing (Day N, N+1, N+3, N+7, and N+14). Participants were assigned to one of two groups: the mixture group (n = 12), trained and tested with binary odor mixtures, or the enantiomer group (n = 12), trained and tested with odor enantiomers. Each participant trained with a designated nostril (trained nostril) and a specific odor pair (training pair). (**B**) Chemical structures of the olfactory stimuli. Top: constituents of two binary odor mixture pairs; bottom: two enantiomer pairs. (**C**) Improvements in mixture discrimination (red) and enantiomer discrimination (green) over the course of training. Data points were interpolated (gridded interpolation) and averaged across participants; squares indicate mean discrimination accuracies for the training pair in the trained nostril at baseline and at the Day N post-training test. (**D**) Discrimination accuracies at baseline (Day 0, lighter bars) and at the Day N post-training test (darker bars) for the training pair (red and green bars) and the control pair (brown and blue bars), presented to the trained (solid bars) and untrained (striped bars) nostrils in the mixture group (top) and enantiomer group (bottom). Gray lines represent individual participants. (**E**) Discrimination accuracies across post-training test sessions for the training and control pairs, presented to the trained and untrained nostrils in the mixture group (top) and enantiomer group (bottom). Black dashed lines: chance level (1/3). Shaded areas and error bars: SEMs. **p < 0.01, ***p ≤ 0.001.

The online version of this article includes the following figure supplement(s) for figure 1:

**Figure supplement 1.** Odor evaluations of guaiacol, eugenol, 2-butanol, and 2-heptanol by a panel of 24 participants.

**Figure supplement 2.** Experiment 1: training-related olfactory adaptation and recovery.

participants received training with the phenolic mixtures, using the alcoholic mixtures as controls; the remaining seven received the reverse assignment. In parallel, the enantiomer group was presented with two pairs of structurally unrelated odor enantiomers: the enantiomers of α-pinene and those of 2-butanol (*Figure 1B*; *Feng and Zhou, 2019*). Each pair served as the training pair for half of the participants and as the control pair for the other half.

Olfactory discrimination was assessed unilaterally using a triangle task. Participants, blindfolded and using one nostril (the other nostril pinched shut), sniffed three bottles in sequence. Two bottles contained the same odorant, and one contained a different odorant; participants identified the odd odor (chance = 1/3). Feedback was provided during training (*Roelfsema et al., 2010*) but not during baseline (Day 0) or post-training testing. Training was conducted in the left nostril for roughly half of the participants in each group and in the right nostril for the other half. Each daily session (Day 1 to Day N) comprised 12 trials with only the training pair presented. Training continued until discrimination accuracy exceeded 83% (≥ 10/12 correct) on two consecutive days (Day N-1 and Day N).

Post-training testing occurred 1 hour after the final training session (Day N) and again after 1, 3, 7, and 14 days (Day N+1, N+3, N+7, and N+14). At baseline and post-training, both training and control pairs were tested in both nostrils (nine trials per pair per nostril per session).

At baseline, both groups performed at chance (mixture: 120 correct out of 36×12 = 432 trials, binomial test p = 0.99; enantiomer: 152/432 correct, binomial test p = 0.22), regardless of odor pair identity (phenolic vs. alcoholic: $F_{1, 11}$ = 1.92, p = 0.19; α-pinene vs. 2-butanol: $F_{1, 11}$ = 0.88, p = 0.37) or nostril (left vs. right: $Fs_{1, 11}$ = 0.084 and 0.77, ps = 0.78 and 0.40). Training sessions to criterion ranged from 7 to 28 in both groups, with no group difference ($t_{22}$ = 1.32, p = 0.20) and no sex effect ($t_{22}$ = 1.46, p = 0.16). Learning curves showed similar continuous improvements between groups, indicating retention across days (*Karni and Sagi, 1993*; *Figure 1C*). Thus, despite interindividual variability, mixture and enantiomer discrimination tasks were comparable in difficulty.

An omnibus ANOVA on Day N post-training discrimination accuracies, with odor pair (training vs. control) and nostril (trained vs. untrained) as within-subjects factors and group (mixture vs. enantiomer) as the between-subjects factor, revealed a robust three-way interaction ($F_{1, 22}$ = 19.96, p < 0.001, $\eta^2_p$ = 0.48). Both groups showed significant perceptual gains for the training pair in the trained nostril (from 0.32 at baseline to 0.76 at post-training, $t_{23}$ = 10.93, p = 1.40 × $10^{-10}$, Cohen's d = 2.23), with no group or sex difference (group: $t_{22}$ = –0.45, p = 0.66; sex: $t_{22}$ = 0.22, p = 0.83). However, transfer patterns diverged markedly between groups (*Figure 1D*).

In the mixture group, significant improvements were also observed for the control pair in the trained nostril, as well as for both the training and control pairs in the untrained nostril ($ts_{11}$ = 5.90, 6.29, and 3.51, ps < 0.001, < 0.001, and = 0.005, Cohen's ds = 1.70, 1.82, and 1.01). Gains were smaller for the control pair than for the training pair (0.22 vs. 0.40; odor pair: $F_{1, 11}$ = 25.88, p < 0.001, $\eta^2_p$ = 0.70), but did not differ by nostril (nostril: $F_{1, 11}$ = 1.77, p = 0.21; odor pair × nostril: $F_{1, 11}$ = 0.20, p = 0.66). Overall, participants performed above chance for both training and control mixtures (0.70 and 0.48, $ts_{11}$ = 9.69 and 3.22, ps < 0.001 and = 0.008, Cohen's ds = 2.80 and 0.93), independent of nostril ($ts_{11}$ = 1.47 and 1.00, ps = 0.17 and 0.34). Thus, mixture discrimination learning transferred fully across nostrils and generalized to untrained mixtures differing in structure and odor quality.

In the enantiomer group, learning was confined to the training pair in the trained nostril. No significant gains were observed for the control pair in the trained nostril or for either pair in the untrained nostril ($ts_{11}$ = 0.80, 0.83, and 1.96, ps = 0.44, 0.42, and 0.076), consistent with prior findings on chiral discrimination learning (*Feng and Zhou, 2019*) and indicative of nostril- and structure-specific plasticity.

Having established these immediate post-training effects, we next asked whether they were maintained over time. Separate repeated-measures ANOVAs were conducted on post-training discrimination accuracies for each group, with odor pair (training vs. control), nostril (trained vs. untrained), and test session (Day N, N+1, N+3, N+7, and N+14) as within-subjects factors. In both groups, there was no significant main effect of test session (mixture: $F_{4, 44}$ = 0.33, p = 0.86; enantiomer: $F_{4, 44}$ = 2.48, p = 0.058) and no interaction between test session and the other factors (ps > 0.1). In the mixture group, a significant main effect of odor pair was observed ($F_{1, 11}$ = 14.95, p = 0.003, $\eta^2_p$ = 0.58), with no effect of nostril or odor pair × nostril interaction ($Fs_{1, 11}$ = 0.53 and 0.001, ps = 0.48 and 0.97). By contrast, in the enantiomer group, there was a significant odor pair × nostril interaction ($F_{1, 11}$ = 22.92, p < 0.001, $\eta^2_p$ = 0.68), along with significant main effects of odor pair and nostril ($Fs_{1, 11}$ = 19.18 and 32.81, ps = 0.001 and < 0.001, $\eta^2_p$s = 0.64 and 0.75). Thus, discrimination performances remained stable in both groups, and their respective patterns of specificity and transfer were preserved across all test sessions (*Figure 1E*). Two weeks post-training (Day N+14), participants in the mixture group continued to discriminate both the training and control pairs of odor mixtures significantly above chance ($ts_{11}$ = 5.26 and 4.37, ps < 0.001 and = 0.001, Cohen's ds = 1.52 and 1.26), regardless of nostril ($ts_{11}$ = 0.29 and 0.28, ps = 0.78 and 0.78). In contrast, participants in the enantiomer group still showed supra-chance performance only for the training pair presented to the trained nostril ($ts_{11}$ = 5.45, p < 0.001, Cohen's d = 1.57; ps > 0.25 for all other conditions).

Beyond discrimination accuracy, prolonged unilateral olfactory training also induced odor- and nostril-specific adaptation effects. At baseline, participants rated the training and control pairs presented to each nostril as similarly intense (mixture: $F_{3, 33}$ = 0.30, p = 0.83; enantiomer: $F_{3, 33}$ = 0.13, p = 0.94). At the post-training test on Day N, however, the training pair presented to the trained nostril was rated as significantly weaker than the same pair presented to the untrained nostril, as well

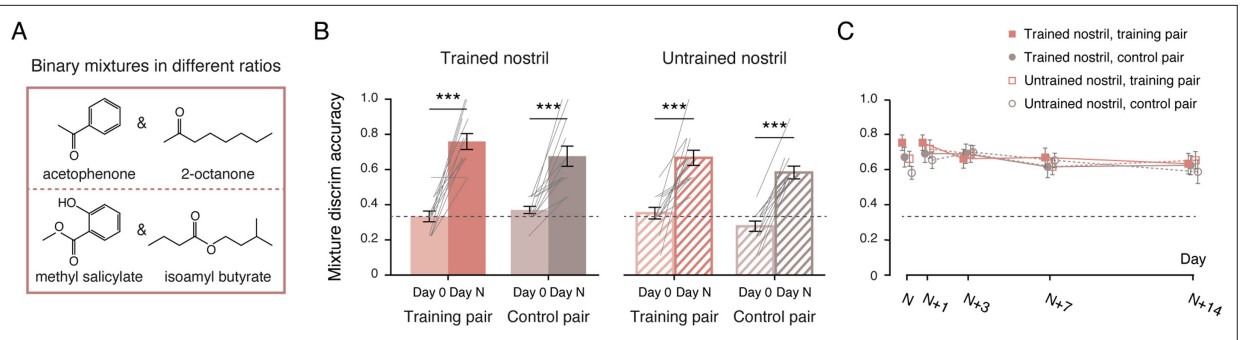

**Figure 2.** Experiment 2: replication and extension of long-term transfer and generalization of mixture discrimination learning. (**A**) Chemical structures of the constituents of two new binary odor mixture pairs used in Experiment 2. (**B**) Discrimination accuracies at baseline (Day 0, lighter bars) and at the Day N post-training test (darker bars) for the training pair (red bars) and the control pair (brown bars), presented to the trained (solid bars) and untrained (striped bars) nostrils. Gray lines represent individual participants (n = 12). (**C**) Discrimination accuracies across post-training test sessions (Day N, N+1, N+3, N+7, and N+14) for the training and control pairs, presented to the trained and untrained nostrils. Black dashed lines: chance level (1/3). Error bars: SEMs. ***$p \leq 0.001$.

The online version of this article includes the following figure supplement(s) for figure 2:

**Figure supplement 1.** Odor evaluations of acetophenone, 2-octanone, methyl salicylate, and isoamyl butyrate by a new panel of 24 participants.

**Figure supplement 2.** Experiment 2: training-related olfactory adaptation and recovery.

as the control pairs presented to either nostril (ps < 0.01, **Figure 1—figure supplement 2A**). No differences were observed among the latter three conditions ($F_{2, 22}$ = 0.43 and 0.048, ps = 0.66 and 0.95). This adaptation effect diminished gradually over time ($F_{5.18, 62.16}$ = 3.81, p = 0.004, $\eta^2_p$ = 0.24, **Figure 1—figure supplement 2B**). By 2 weeks post-training (Day N+14), perceived odor intensities no longer differed by nostril (training pair: $t_{15}$ = –0.55, p = 0.59; control pair: $t_{15}$ = 1.38, p = 0.19), while discrimination abilities remained robust (**Figure 1E**).

## Replication and extension of mixture learning generalization and persistence

Experiment 1 demonstrated that mixture discrimination learning generalized across nostrils to untrained mixtures and persisted over time, in contrast to the nostril- and structure-specificity of enantiomer discrimination learning. One possible explanation for the observed transfer between phenolic and alcoholic mixtures is that both share a common structural feature—the hydroxyl group. To test whether transfer of mixture learning requires such a shared functional group, we introduced two new pairs of odor mixtures in Experiment 2 (**Figure 2A**). One pair consisted of a:b and b:a mixtures of acetophenone (1% v/v in propylene glycol) and 2-octanone (1% v/v), and the other of a:b and b:a mixtures of methyl salicylate (1% v/v) and isoamyl butyrate (1% v/v), with no shared functional group between the two sets. The constituents of each binary mixture, one aromatic and the other aliphatic, were also structurally more distinct than those in Experiment 1. This experiment, therefore, served both to replicate the mixture learning effect observed in Experiment 1 and to extend it to novel structurally unrelated stimulus sets.

Based on ratings from a new panel of 24 odor judges (**Figure 2—figure supplement 1A**), acetophenone and 2-octanone were comparable in odor intensity and valence ($ts_{23}$ = –1.56 and 0.30, ps = 0.13 and 0.76), whereas methyl salicylate was equally intense to but less pleasant than isoamyl butyrate ($ts_{23}$ = –1.07 and –2.64, ps = 0.29 and 0.015). All compounds were odor-dissimilar (mean pairwise similarity ratings < 3.2 for any two compounds), with no difference in perceptual similarity between those within the same functional group (both ketones or both esters, 2.5 ± 1.3) and those in different functional groups (one ketone, one ester, 2.5 ± 0.8; $t_{23}$ = 0.45, p = 0.65; **Figure 2—figure supplement 1B**). Using the same procedures as in Experiment 1 (**Figure 1A**), we trained and tested 12 participants unilaterally. The mixing ratio a:b was 9:11 for nine participants and 7:9 for three participants.

At baseline, participants performed at chance in discriminating the binary mixtures (144/432 correct, binomial test p = 0.52), regardless of mixture type (ketones vs. esters: $F_{1, 11}$ = 0, p > 0.99) or

nostril (left vs. right: $F_{1, 11}$ = 0.077, p = 0.79). Participants were then randomly assigned to training in either the left nostril (n = 5) or the right nostril (n = 7), with either the ketone mixtures (n = 7; ester mixtures as control) or the ester mixtures (n = 5; ketone mixtures as control). Training required 16–34 sessions (N = 16 to N = 34) to reach criterion, with no sex difference (Mann–Whitney Z = –0.56, p = 0.57).

Comparisons of baseline and post-training performance on Day N revealed substantial improvements not only for the training pair in the trained nostril ($t_{11}$ = 7.37, p < 0.001, Cohen's d = 2.13), but also for the control pair in the trained nostril, the training pair in the untrained nostril, and the control pair in the untrained nostril ($ts_{11}$ = 5.25, 5.17, and 7.84, ps < 0.001, Cohen's ds = 1.51, 1.49, and 2.26; *Figure 2B*). These effects were comparable across sexes (Mann–Whitney |Z|s < 1.46, ps > 0.14). Specifically, participants trained with the ketone mixtures discriminated both ketone and ester mixtures above chance in both nostrils (mean accuracies = 0.75, 0.71, 0.65, and 0.52; Wilcoxon Zs = 2.37, 2.38, 2.38, and 2.53; ps = 0.018, 0.017, 0.017, and 0.011). Similarly, those trained with the ester mixtures discriminated both ester and ketone mixtures above chance in both nostrils (mean accuracies = 0.78, 0.60, 0.71, and 0.67; Wilcoxon Zs = 2.06, 2.03, 2.02, and 2.04; ps = 0.039, 0.042, 0.043, and 0.041). Thus, the learning effect transferred across nostrils and between ketone and ester mixtures that were perceptually dissimilar and shared no functional group, echoing the results obtained with phenolic and alcoholic mixtures in Experiment 1. Moreover, discrimination accuracies for the untrained mixtures exceeded those in Experiment 1 by 15% (0.63 vs. 0.48, $t_{22}$ = 2.55, p = 0.018, Cohen's d = 1.04), suggesting that structural diversity of the training material may enhance generalization. Training also induced a nostril-specific adaptation effect: the training pair presented to the trained nostril came to be perceived as significantly less intense than the same pair in the untrained nostril, as well as the control pairs in either nostril (ps ≤ 0.043), whereas no such differences were observed at baseline ($F_{3, 33}$ = 0.60, p = 0.62; *Figure 2—figure supplement 2A*).

As in Experiment 1, discrimination performance and transfer patterns remained stable across the five post-training test sessions spanning 2 weeks (Day N, N+1, N+3, N+7, and N+14, $F_{4, 44}$ = 1.42, p = 0.24, interactions: ps > 0.1; *Figure 2C*). Although discrimination accuracies were numerically lower for the control pair than for the training pair, this difference was not significant ($F_{1, 11}$ = 3.50, p = 0.088), reflecting substantial generalization. Performances were comparable between trained and untrained nostrils (nostril: $F_{1, 11}$ = 2.16, p = 0.17; odor pair × nostril: $F_{1, 11}$ = 0.019, p = 0.89), indicating complete transfer across nostrils. Meanwhile, training-induced adaptation diminished gradually ($F_{12, 132}$ = 2.75, p = 0.002, $\eta^2_p$ = 0.20, *Figure 2—figure supplement 2B*). Two weeks post-training (Day N+14), participants continued to discriminate both the training and control pairs significantly above chance ($ts_{11}$ = 6.30 and 4.93, ps < 0.001, Cohen's ds = 1.82 and 1.42), irrespective of nostril ($ts_{11}$ = –0.39 and 0.74, ps = 0.70 and 0.47). By this time, perceived odor intensities no longer differed by nostril or whether the mixtures had been used for training (main effects: $Fs_{1, 11}$ = 0.60 and 0.026, ps = 0.46 and 0.87; interaction: $F_{1, 11}$ = 0.061, p = 0.81). Thus, Experiment 2 confirmed that mixture discrimination learning generalized robustly across nostrils and to structurally distinct odor mixtures lacking a common functional group, and that these effects persisted over time.

## Dissociating mixture from concentration discrimination learning

In mixture discrimination learning, participants were trained to distinguish between a:b and b:a mixtures of two compounds. It could be argued, however, that participants were not truly learning to discriminate the configural perceptual qualities of the mixtures; rather, the task may have reduced to discriminating between relative concentrations of the individual components. To test this possibility, Experiment 3 examined concentration discrimination learning directly, using the constituents of the mixtures from Experiment 2 (*Figure 2A*) as stimuli. For each compound, two solutions of different concentrations (0.44% and 0.56% v/v, corresponding to a 7:9 ratio) were prepared, yielding four concentration pairs: acetophenone, 2-octanone, methyl salicylate, and isoamyl butyrate. Twelve participants were then trained and tested with these concentration pairs, following the same procedures as in the previous experiments. Half were randomly assigned to receive training with the acetophenone concentration pair, with 2-octanone and isoamyl butyrate serving as structurally related (sharing a carbonyl group) and structurally-unrelated controls, respectively; the other half were trained with the 2-octanone concentration pair, with acetophenone and methyl salicylate as structurally related and

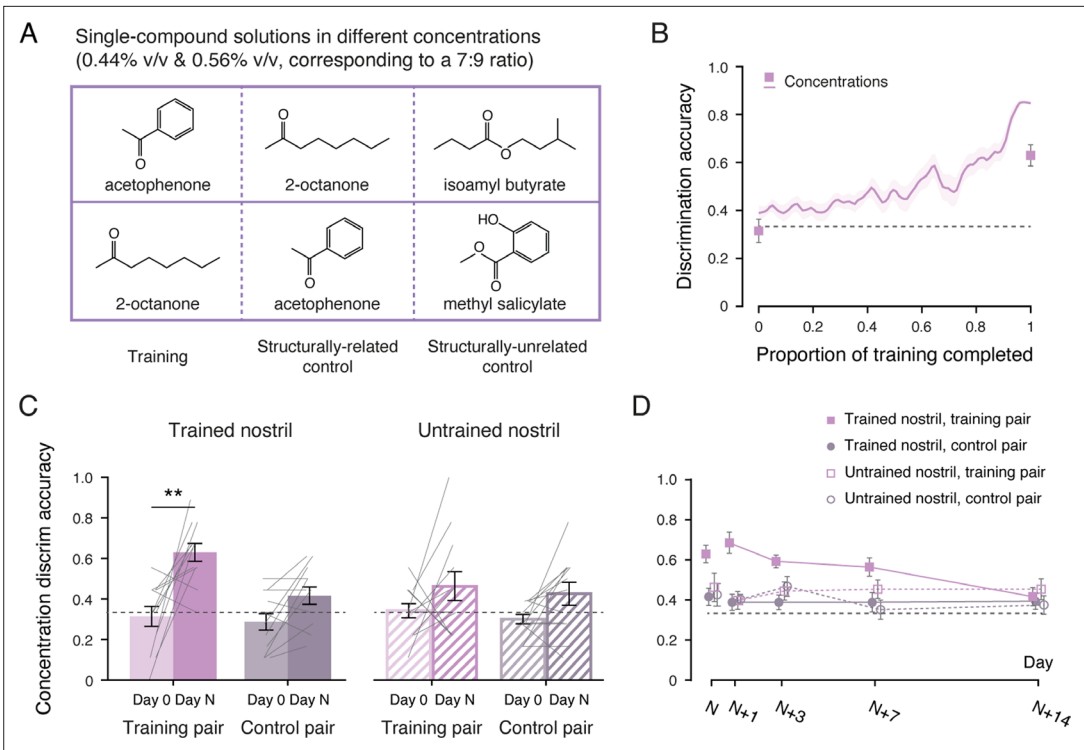

**Figure 3.** Experiment 3: specificity and transience of concentration discrimination learning. (**A**) Chemical structures of the single-compound solutes used for the training and control solutions in Experiment 3. (**B**) Improvements in concentration discrimination over the course of training. Data points were interpolated (gridded interpolation) and averaged across participants (n = 12); squares indicate mean discrimination accuracies for the training pair in the trained nostril at baseline and at the Day N post-training test. (**C**) Concentration discrimination accuracies at baseline (Day 0, lighter bars) and at the Day N post-training test (darker bars) for the training pair (purple bars) and the control pair (gray bars), presented to the trained (solid bars) and untrained (striped bars) nostrils. Gray lines represent individual participants. (**D**) Concentration discrimination accuracies across post-training test sessions (Day N, N+1, N+3, N+7, and N+14) for the training and control pairs, presented to the trained and untrained nostrils. Black dashed lines: chance level (1/3). Error bars: SEMs. **p < 0.01.

The online version of this article includes the following figure supplement(s) for figure 3:

**Figure supplement 1.** Experiment 3: concentration discrimination and training-related olfactory adaptation and recovery.

structurally unrelated controls, respectively (*Figure 3A*). Training was conducted unilaterally, in the left nostril for half of the participants and in the right nostril for the remainder.

Participants performed at chance for all concentration pairs at baseline (discrimination accuracies < 0.33, binomial tests ps > 0.6), irrespective of nostril (left vs. right: $t_{11}$ = –0.50, p = 0.63). Training required 14–38 sessions to reach criterion, comparable to training with enantiomers and mixtures in Experiments 1 and 2 ($F_{2, 45}$ = 1.85, p = 0.17), with no sex difference (Mann–Whitney Z = –1.64, p = 0.10). Comparison of the learning curves across concentration, enantiomer, and mixture discrimination learning indicated slower acquisition and more limited retention during the early portion of training for concentration discrimination, followed by steeper improvement in the later sessions (*Figure 3B*, *Figure 3—figure supplement 1A*). Several participants also self-reported adopting cognitive strategies in the later sessions, such as weighing an odor's perceived intensity against its serial position (first, second, or third) within a trial.

*Figure 3C* contrasts post-training performance on Day N with baseline. The structurally related and unrelated control pairs were grouped together, as they exhibited comparable post-training accuracies (*Figure 3—figure supplement 1B*), with no effect of structural relatedness (related vs. unrelated: $F_{1, 11}$ = 2.05, p = 0.18), nostril (trained vs. untrained: $F_{1, 11}$ = 0.066, p = 0.80), or their interaction ($F_{1, 11}$ = 0.22, p = 0.65). Post-training concentration discrimination was significantly above chance for the

training pair in the trained nostril ($t_{11}$ = 6.74, p < 0.001, Cohen's d = 1.95), representing a significant improvement over baseline ($t_{11}$ = 3.85, p = 0.003, Cohen's d = 1.11), with no effect of sex (Mann–Whitney Z = –1.40, p = 0.16). By contrast, performance did not differ from chance for the same pair in the untrained nostril or for the control pairs in either nostril ($ts_{11}$ = 1.83, 1.94, and 1.65, ps = 0.095, 0.079, and 0.13). This pattern was distinct from that observed for mixture discrimination learning (*Figures 1B and 2B*) and more similar to that for enantiomer discrimination learning (*Figure 1B*), suggestive of the involvement of peripheral olfactory mechanisms. Immediate post-training accuracy for the training pair in the trained nostril was lower for concentration discrimination than for either enantiomer ($t_{22}$ = –2.22, p = 0.037, Cohen's d = 0.91) or mixture discrimination ($t_{34}$ = 2.12, p = 0.041, Cohen's d = 0.75), despite the identical training procedure and criterion and comparable chance-level accuracies at baseline (ps > 0.7), pointing to a less robust learning effect. This was paralleled by odor- and nostril-specific adaptation (*Figure 3—figure supplement 1C*), as in the previous experiments. At the Day N post-training test, the training pair in the trained nostril was rated as significantly weaker than the other combinations of odor pair (training, structurally related control, and structurally unrelated control) and nostril (trained vs. untrained) (ps ≤ 0.017), with no difference among the latter ($F_{4, 44}$ = 0.70, p = 0.59). All combinations were rated as equally intense at baseline ($F_{2.9, 31.7}$ = 1.37, p = 0.27). For all odor pairs, there was no significant change in perceived valence from baseline to Day N, regardless of nostril (ps > 0.05 for the main effects of session and nostril, as well as their interaction; *Figure 3—figure supplement 1D*).

Unlike enantiomer or mixture discrimination learning, however, the effect of concentration discrimination learning was short-lived (*Figure 3D*). Repeated-measures ANOVAs on discrimination accuracies across the five post-training test sessions (Day N, N+1, N+3, N+7, and N+14) revealed a significant three-way interaction among odor pair (training vs. control), nostril (trained vs. untrained), and test session ($F_{4, 44}$ = 3.58, p = 0.013, $\eta^2_p$ = 0.25), along with significant effects of odor pair and nostril, as well as their interaction ($Fs_{1, 11}$ = 6.17, 11.40, and 17.00, ps = 0.030, 0.006, and 0.002, $\eta^2_p$ = 0.36, 0.51, and 0.61). These results indicated that discrimination performance, and its pattern of specificity, changed over the course of follow-up testing. This change was driven by a monotonic decline in accuracy for the training pair in the trained nostril ($F_{4, 44}$ = 7.26, p < 0.001, $\eta^2_p$ = 0.40), which appeared to parallel recovery from the odor- and nostril-specific adaptation (*Figure 1—figure supplement 2B*, *Figure 2—figure supplement 2B*, and *Figure 3—figure supplement 1E*). Odor valence ratings remained stable across sessions (ps ≥ 0.29 for the main and interaction effects involving session), showing the same pattern as at baseline (*Figure 3—figure supplement 1D and F*). By 2 weeks post-training (Day N+14), accuracy for the training pair in the trained nostril had dropped to chance level ($t_{11}$ = 1.83, p = 0.095), and neither concentration discrimination performance nor odor intensity ratings differed by nostril or by whether the concentration pair had been used for training (nostril: $Fs_{1, 11}$ = 0.040 and 0.036, ps = 0.84 and 0.85; odor pair: $Fs_{1, 11}$ = 1.09 and 0.79, ps = 0.32 and 0.39; interaction: $Fs_{1, 11}$ = 0.33 and 2.45, ps = 0.58 and 0.15).

Taken together, these findings suggest that concentration discrimination learning could reflect the acquisition of strategies hinging on peripheral adaptation and that such learning dissipates within 2 weeks after training. Importantly, concentration discrimination learning cannot account for the generalization and persistence observed in mixture discrimination learning; instead, these two forms of olfactory learning are supported by distinct mechanisms.

## Discussion

Generalization enables the transfer of learning from specific parameters to broader, novel situations. This typically involves the abstraction of knowledge or characteristic patterns from specific circumstances (*Kellman and Massey, 2013*; *Tenenbaum et al., 2011*). In this study, we employed a common protocol but introduced different types of odors for unilateral olfactory discrimination training, revealing contrasting patterns of specificity, transfer, and persistence. Learning was confined to the training pair presented to the trained nostril in participants trained with odor enantiomers or with different concentrations of a single compound. By contrast, in those trained with odor mixtures, learning transferred completely to the untrained nostril and generalized—though not fully—to structurally and perceptually unrelated mixtures. The structural diversity within the training mixtures appeared to further enhance this generalization. These results emerged despite highly similar subjective experiences—discriminating between two highly similar odors—and

comparable task difficulties across groups. Moreover, the learning effects were long-lasting for both enantiomers and mixtures, persisting for at least 2 weeks, and were not dependent on olfactory adaptation or sex, consistent with long-term plasticity of the human olfactory system. In contrast, concentration discrimination learning was short-lived and appeared to depend on peripheral adaptation.

To interpret these behavioral patterns, we consider the anatomical organization of the olfactory system. Olfactory inputs to the two nostrils remain largely segregated up to the primary olfactory cortex, where the piriform cortex, the largest structure, serves as the principal recipient of afferents from the olfactory bulb (*Gottfried and Zald, 2005*). The piriform cortex is known to transform the chemical feature space inherited from the bulb into an odor quality space characterized by sparse coding (*Bekkers and Suzuki, 2013*; *Howard et al., 2009*; *Pashkovski et al., 2020*; *Stettler and Axel, 2009*). Downstream projections from the primary olfactory cortex to the orbitofrontal cortex, lateral and basolateral amygdala, hippocampus, and other limbic and paralimbic structures link olfactory inputs to systems mediating affect, learning, memory, and value appraisal (*Gottfried, 2010*; *Gottfried and Zald, 2005*). The nostril- and structure-specificity of enantiomer and concentration discrimination learning likely reflects modifications in or upstream of the olfactory bulb (*Feng and Zhou, 2019*). Conversely, the inter-nostril transfer of mixture discrimination learning suggests that learning occurs at a stage where binaral inputs converge, that is, beyond the primary olfactory cortex. The generalization of mixture discrimination learning to unrelated mixtures further indicates the involvement of abstract representations of discrimination boundaries for inputs varying in odor quality. A plausible locus of plasticity underlying such generalization is the orbitofrontal cortex, the principal neocortical target of primary olfactory areas (*Gottfried, 2010*). This region is involved in the perceptual coding of odor identity (*Li et al., 2006*), valence (*Anderson et al., 2003*), and other properties and plays a key role in olfactory discrimination (*Li et al., 2006*; *Zatorre and Jones-Gotman, 1991*). In particular, it has been proposed to integrate olfactory perceptual evidence toward a decision criterion in mixture discrimination (*Bowman et al., 2012*). Prolonged training with odor mixtures may refine the representation of perceptual evidence (e.g., relative differences in configural odor notes) in the orbitofrontal cortex and modify the discrimination boundary (*Qamar et al., 2013*). Applying this optimized boundary could yield generalized improvements in mixture discrimination beyond the trained nostril and training pair.

The observed differences in specificity, transfer, and persistence between mixture discrimination learning and enantiomer or concentration discrimination learning cannot be explained by generic factors such as attention, adaptation, reinforcement, task difficulty, or training duration (*Ahissar and Hochstein, 1997*; *Ahissar and Hochstein, 2004*; *Harris et al., 2012*; *Karni and Sagi, 1993*; *Roelfsema et al., 2010*). Instead, they align with the reweighting model of perceptual learning (*Lu and Dosher, 2022*) and the broader notion that learning relies on distributed plasticity across the brain (*Maniglia and Seitz, 2018*). The weight structure acquired for discriminating odor mixtures (e.g., relative differences in configural odor notes) may approximate that required for other mixtures, thereby enabling transfer. By contrast, the weight structures for enantiomer or concentration discrimination (e.g., chiral configuration or perceived intensity relative to serial position) are too specific to generalize, resulting in high specificity. For concentration discrimination, the weight structure likely depends on nostril- and odor-specific adaptation, and diminishes as adaptation recovers. In simpler terms, learning to discriminate different odors engages plasticity at distinct stages of olfactory processing.

These findings bear important practical implications. They demonstrate the generalizability (with mixture discrimination) and persistence (with both enantiomer and mixture discrimination, but not concentration discrimination) of learning-induced olfactory gains, supporting the potential value of olfactory training as both a therapeutic intervention for olfactory dysfunction and as a means of acquiring olfactory expertise. Critically, they highlight the need to employ complex odors and active tasks, rather than single compounds or passive exposure, to promote long-term generalization of learning effects beyond the training context. Given the anatomical proximity of secondary olfactory areas to regions implicated in emotion and memory (*Gottfried, 2010*; *Gottfried and Zald, 2005*), our results also raise the intriguing possibility that olfactory training, when implemented with carefully designed materials and protocols, may influence emotional and mnemonic functions. This possibility warrants systematic investigation in future studies.

## Materials and methods

### Participants

A total of 96 healthy, non-smoking individuals participated. 48 completed unilateral odor discrimination training (average duration = 20.0 days) together with baseline and post-training testing: 24 (12 females; 22.3 ± 1.4 years) in Experiment 1, 12 (6 females; 25.1 ± 1.5 years) in Experiment 2, and 12 (6 females; 25.9 ± 2.1 years) in Experiment 3. In total, these participants completed 1247 training and testing sessions. Group sizes (n = 12) were consistent with prior perceptual learning studies (*Dalton et al., 2002*; *Feng and Zhou, 2019*; *Mainland et al., 2002*). The remaining 48 participants served as odor judges: 24 (12 females; 25.3 ± 2.3 years) rated the mixture constituents used in Experiment 1, and 24 (12 females; 25.2 ± 2.2 years) rated those used in Experiments 2 and 3. All participants reported a normal sense of smell and no respiratory allergies or infections at the time of testing. They provided written informed consent to participate in procedures approved by the Institutional Review Board at the Institute of Psychology, Chinese Academy of Sciences (H18029).

### Olfactory stimuli

In Experiment 1, the olfactory stimuli comprised two pairs of binary mixtures and two pairs of enantiomers (*Figure 1B*): a:b and b:a mixtures of guaiacol (CAS 90-05-1, 1% v/v) and eugenol (97-53-0, 1% v/v); a:b and b:a mixtures of 2-butanol (78-92-2, 1% v/v) and 2-heptanol (543-49-7, 1% v/v); (+)-α-pinene (7785-70-8, 1% v/v) and (-)-α-pinene (7785-26-4, 1% v/v); and (+)–2-butanol (4221-99-2, 1% v/v) and (-)–2-butanol (14898-79-4, 1% v/v). In Experiment 2, the stimuli consisted of two new pairs of binary mixtures (*Figure 2A*): a:b and b:a mixtures of acetophenone (98-86-2, 1% v/v) and 2-octanone (111-13-7, 1% v/v), and a:b and b:a mixtures of methyl salicylate (119-36-8, 1% v/v) and isoamyl butyrate (106-27-4, 1% v/v). The mixing ratio a:b was 9:11 for three participants in Experiment 1 and nine participants in Experiment 2, and 7:9 for nine participants in Experiment 1 and three participants in Experiment 2. These ratios were selected based on pilot testing and practical constraints. Specifically, they met two criteria: baseline indiscriminability (most participants were unable to reliably discriminate between the two binary mixtures in a:b and b:a ratios at baseline) and trainability (with 1–5 weeks of training, participants could acquire the ability to discriminate between them). The mixture constituents used in Experiment 2 also served as the stimuli in Experiment 3 (*Figure 3A*). For each compound, two concentrations (0.44% and 0.56% v/v, corresponding to a 7:9 ratio) were prepared, yielding four concentration pairs: acetophenone, 2-octanone, methyl salicylate, and isoamyl butyrate. All compounds were dissolved in propylene glycol. Mixture constituents were evaluated for odor characteristics prior to the main experiments.

Stimuli were presented in identical 40 ml polypropylene jars, each containing 10 ml of solution. Jars were fitted with either one Teflon nosepiece (for unilateral training and testing) or two nosepieces (for odor evaluations). Participants were instructed to inhale through the nosepieces and exhale through the mouth. Fresh odor solutions were prepared every other day during data collection.

### Procedure

#### Olfactory discrimination training and testing

Procedures were adapted from a previous study on olfactory perceptual learning of chiral discrimination (*Feng and Zhou, 2019*). Experiment 1 comprised three phases: baseline, unilateral training, and post-training testing (*Figure 1A*).

At baseline (Day 0), blindfolded participants completed a unilateral odor intensity rating task followed by a unilateral odor discrimination task. They were instructed to keep one nostril pinched shut for the duration of each trial, while odors were presented to the open nostril by the experimenter. For the mixture group, stimuli were two pairs of binary odor mixtures (guaiacol/eugenol and 2-butanol/2-heptanol, in both a:b and b:a ratios). For the enantiomer group, stimuli were the enantiomers of α-pinene and of 2-butanol. In the odor intensity rating task, participants were presented with one odor per trial and rated its intensity on a 7-point Likert scale (1 = not at all intense; 7 = extremely intense). The two odors within a pair were presented in random order across four consecutive trials, with nostril of presentation alternated (one trial per odor per nostril). In the odor discrimination task, participants were presented with three sequential odors per trial (two identical, one different) and asked to identify the odd stimulus. Each odor pair was tested in 18 consecutive trials, with nostril of

presentation alternated (nine trials per pair per nostril). Odor pair order was counterbalanced across participants. A 30 s break was given between trials, and in the odor discrimination task, a 10 min break was provided after the first 18 trials. No feedback was given. Participants with discrimination accuracy ≤ 5/9 (chance = 1/3) for each odor pair were invited to training.

Training began the following day (Day 1) and was conducted daily at approximately the same time until criterion was reached (Day 1 to Day N). Each session comprised 12 unilateral odor discrimination trials with a designated nostril (trained nostril) and odor pair (training pair), with a 30 s break between trials. Immediate feedback was provided. In the mixture group, five participants trained with the phenolic mixtures and seven with the alcoholic mixtures. In the enantiomer group, six participants trained with α-pinene enantiomers and six with 2-butanol enantiomers. Training was conducted in the left nostril for seven mixture-group participants and six enantiomer-group participants, and in the right nostril for the remainder. Training concluded when discrimination accuracy reached ≥ 10/12 on two consecutive days (Day N-1 and Day N).

The first post-training test was administered one hour after the final training session on Day N, using the same stimuli and procedures as at baseline. Retests were conducted 1, 3, 7, and 14 days later (Day N+1, N+3, N+7, and N+14). These sessions followed the baseline and Day N post-training test procedures, except that not all participants completed the unilateral odor intensity rating task. On Days N+1, N+3, and N+7, ratings were provided by nine mixture-group participants and four enantiomer-group participants; on Day N+14, ratings were provided by 12 mixture-group participants and four enantiomer-group participants.

Apart from the stimuli used, procedures in Experiments 2 and 3 were identical to those in Experiment 1. In Experiment 2, seven participants trained with the ketone mixtures, with the ester mixtures serving as the control condition, while the remaining five received the opposite assignment. Training was conducted in the left nostril for five participants and in the right nostril for seven. In Experiment 3, six participants trained with the acetophenone concentration pair, with 2-octanone and isoamyl butyrate serving as structurally related and structurally unrelated controls. The remaining six trained with the 2-octanone concentration pair, with acetophenone and methyl salicylate as structurally related and structurally unrelated controls. Training was conducted in the left nostril for six participants and in the right nostril for six. All participants in Experiments 2 and 3 completed the unilateral odor intensity rating and odor discrimination tasks at baseline (Day 0) and at the post-training test and retests on Days N, N+1, N+3, N+7, and N+14. Participants in Experiment 3 also provided odor valence ratings on a 7-point Likert scale (1 = not at all pleasant; 7 = extremely pleasant), in addition to odor intensity ratings, during the unilateral odor intensity rating task.

## Odor evaluations of mixture constituents

Two independent panels, each consisting of 24 participants, evaluated the odor characteristics of the mixture constituents used in Experiment 1 (guaiacol, eugenol, 2-butanol, 2-heptanol) and the compounds used in Experiments 2 and 3 (i.e., acetophenone, 2-octanone, methyl salicylate, isoamyl butyrate), respectively. Each participant sampled the compounds individually, in random order, and rated both odor intensity and valence on 7-point Likert scales (1 = not at all intense/pleasant; 7 = extremely intense/pleasant). Subsequently, participants were presented with all six pairwise combinations of the four compounds [C(4, 2)], one pair at a time in random order. For each pair, they rated the perceptual similarity in odor quality of the two odors on a 7-point Likert scale (1 = not at all similar; 7 = extremely similar). A minimum inter-trial interval of 30 s was imposed to reduce olfactory adaptation.

## Statistical analysis

Participants' baseline odor discrimination performance was compared against chance using binomial tests. Sex differences were assessed with independent-samples t-tests and Mann–Whitney U tests (nonparametric, suitable for small sample sizes). Unilateral discrimination accuracies for different odor pairs, both at baseline and at post-training tests, were analyzed using repeated-measures ANOVAs. Specifically, the ANOVAs examined: (1) the effects of odor pair identity and nostril (left vs. right) on baseline discrimination accuracies within each group; (2) the effects of odor pair role (training vs. control) and nostril (trained vs. untrained) on post-training discrimination accuracies on Day N and Day N+14, with group (mixture vs. enantiomer) included as a between-subjects factor in Experiment 1; (3) the effects of post-training test session (Day N, N+1, N+3, N+7, and N+14), odor pair role

(training vs. control), and nostril (trained vs. untrained) on discrimination performances across sessions in each group. In addition, t-tests and Wilcoxon signed-rank tests (non-parametric, suitable for small sample sizes) were conducted to compare discrimination accuracies or training-induced perceptual gains (defined as the difference between Day N post-training and baseline discrimination accuracies) across conditions, and to test discrimination accuracies against chance (1/3). Unilateral odor intensity/valence ratings for different odor pairs were analyzed in a similar manner using repeated-measures ANOVAs and t-tests. In Experiment 1, both groups showed nostril-specific adaptation to the training odor pair at the Day N post-training test. Intensity rating data from the two groups were therefore combined to examine changes in this adaptation effect across sessions (Day N, N+1, N+3, N+7, and N+14).

In the statistical reporting, Fs denote F-values, ts denote t-values (test statistics), and ps denote p values. Effect sizes for ANOVAs and t-tests were estimated using partial eta squared ($\eta^2_p$) and Cohen's d, respectively. All tests were two-tailed where applicable. Reported p values are uncorrected; however, all significant effects remained significant after Holm–Bonferroni correction for multiple comparisons, where applicable.

## Acknowledgements

This work was supported by Brain Science and Brain-like Intelligence Technology – National Science and Technology Major Project 2021ZD0204200, the National Natural Science Foundation of China Grant 32430043, and the Fundamental Research Funds for the Central Universities E4EQ5001X2.

## Additional information

### Funding

| Funder | Grant reference number | Author |
| --- | --- | --- |
| Ministry of Science and Technology of the People's Republic of China | 2021ZD0204200 | Wen Zhou |
| National Natural Science Foundation of China | 32430043 | Wen Zhou |
| Fundamental Research Funds for the Central Universities | E4EQ5001X2 | Wen Zhou |

The funders had no role in study design, data collection and interpretation, or the decision to submit the work for publication.

### Author contributions

Xiaoyue Chang, Data curation, Formal analysis, Investigation, Visualization, Writing – original draft, Project administration, Writing – review and editing; Huibang Tan, Data curation, Investigation; Jiehui Niu, Kaiqi Yuan, Rui Chen, Investigation; Wen Zhou, Conceptualization, Supervision, Funding acquisition, Visualization, Methodology, Writing – original draft, Project administration, Writing – review and editing

### Author ORCIDs

Xiaoyue Chang  https://orcid.org/0000-0001-5777-2069
Huibang Tan  https://orcid.org/0000-0002-3592-5412
Jiehui Niu  https://orcid.org/0009-0008-3451-1833
Kaiqi Yuan  https://orcid.org/0009-0008-7702-0911
Rui Chen  https://orcid.org/0009-0004-4552-5377
Wen Zhou  https://orcid.org/0000-0001-6730-2116

### Ethics

Human subjects: Participants provided written informed consent to participate in procedures approved by the Institutional Review Board at the Institute of Psychology, Chinese Academy of Sciences (H18029).

### Decision letter and Author response

Decision letter https://doi.org/10.7554/eLife.102999.sa1
Author response https://doi.org/10.7554/eLife.102999.sa2

---

## Additional files

### Supplementary files

MDAR checklist

### Data availability

All primary data and analysis scripts are available at: https://doi.org/10.57760/sciencedb.psych.00845.

The following dataset was generated:

| Author(s) | Year | Dataset title | Dataset URL | Database and Identifier |
|---|---|---|---|---|
| Chang X, Zhou W | 2025 | Distinct patterns of human olfactory learning: specificity, transfer, and persistence across odorant types | https://doi.org/10.57760/sciencedb.psych.00845 | Science Data Bank, 10.57760/sciencedb.psych.00845 |

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
