## [Editor Report]

This important and well controlled study explores the specificity of olfactory perceptual learning. In keeping with previous work, the authors found that learning to discriminate between two enantiomers does not generalize across the nostrils or to unrelated enantiomers, whereas learning to discriminate odor mixtures does generalize across the nostrils and to other odor mixtures, with this learning effect persisting over at least two weeks. The evidence presented to support these findings is convincing, and they will be of interest to scientists working on olfactory perception and learning.

---

## [Decision Letter]

**Decision letter after peer review:**

Thank you for submitting your article "Contrasting patterns of specificity and transfer in human odor discrimination learning" for consideration by *eLife*. Your article has been reviewed by 2 peer reviewers, and the evaluation has been overseen by a Reviewing Editor and Andrew King as the Senior Editor.

Essential Revisions:

The study would be greatly strengthened by the addition of insights into why the generalization of learning differs between enantiomers and odor mixtures. Conducting additional experiments on odor detection threshold and discrimination between different odor concentrations, as recommended by reviewer 1, would help to link the findings of the study more closely to previous work and may shed light on this question.

*Reviewer #1 (Recommendations for the authors):*

This study extends a previous study by the same group on the generalization of odor discrimination from one nostril to the other. In their earlier study, the group showed that learning to discriminate between two enantiomers does not generalize across nostrils. This was surprising given the Mainland and Sobel 2001 study that found that detecting androstenone in people who do not detect it can generalize across the two nostrils. In this study, they confirmed their previous results and reported that, unlike enantiomers, learning to discriminate odor mixtures generalizes across nostrils, generalizes to other odor mixtures, and is persistent over at least two weeks.

This interesting and important result extends our knowledge of this phenomenon and will likely steer more research. It may also help develop new training protocols for people with impairments in their sense of smell.

The main weakness of this study is its scope, as it does not provide substantial insight into why the results differ for enantiomers and why training on odor mixtures generalizes to other odor mixtures.

One can suggest several straightforward extensions to the study that will make this study more influential. For example, generalization of odor detection threshold, generalization of perception (similar to androstenone), discriminating between two different odor concentrations. If these are also generalized across nostrils, then one can suggest a hypothesis as to why enantiomers don't. Other results will raise some other exciting hypotheses.

*Reviewer #2 (Recommendations for the authors):*

The manuscript from Chang et al. taps on an important issue in olfactory perceptual plasticity, named the generalization of perceptual learning effect by training using odors. They employed a discrimination training/learning task with either binary odor mixture or odor enantiomers, and tested for post-training effect at several time intervals. Their results showed contrasting patterns of specificity (enantiomers) and transfer (odor mixtures), and the learning effect persisted at 2 weeks post-training. They demonstrated that the effect was independent of task difficulty, olfactory adaptation and gender.

Overall this was a well-controlled study and shows novel results. The strength of the study includes the consideration of odor structure and perceptual (dis)similarity and the control training condition.

I suggest that the authors discuss their work in relation to works on central representation of certain features of odors (valence, identity).

---

## [Author Response]

Essential Revisions:The study would be greatly strengthened by the addition of insights into why the generalization of learning differs between enantiomers and odor mixtures. Conducting additional experiments on odor detection threshold and discrimination between different odor concentrations, as recommended by reviewer 1, would help to link the findings of the study more closely to previous work and may shed light on this question.

Because exposure-induced reductions in detection thresholds and their underlying mechanisms have already been examined in previous studies (Bremner et al., 2003; Mainland et al., 2002; Wang et al., 1993; Yee and Wysocki, 2001), we focused our new data collection on discrimination between different odor concentrations, incorporated as Experiment 3 in the revised manuscript. We have also expanded the Discussion to include revised analyses and interpretations of differences in specificity, transfer, and persistence across mixture, enantiomer, and concentration discrimination learning, and to relate these findings more directly to prior work. Please refer to our responses to Reviewer 1’s Point 1 and Reviewer 2’s Point 3 for additional details.

Reviewer #1 (Recommendations for the authors):This study extends a previous study by the same group on the generalization of odor discrimination from one nostril to the other. In their earlier study, the group showed that learning to discriminate between two enantiomers does not generalize across nostrils. This was surprising given the Mainland and Sobel 2001 study that found that detecting androstenone in people who do not detect it can generalize across the two nostrils. In this study, they confirmed their previous results and reported that, unlike enantiomers, learning to discriminate odor mixtures generalizes across nostrils, generalizes to other odor mixtures, and is persistent over at least two weeks.This interesting and important result extends our knowledge of this phenomenon and will likely steer more research. It may also help develop new training protocols for people with impairments in their sense of smell.

We thank the reviewer for the encouraging remarks.

The main weakness of this study is its scope, as it does not provide substantial insight into why the results differ for enantiomers and why training on odor mixtures generalizes to other odor mixtures.

We thank the reviewer for this insightful comment. While the present study does not directly identify the neural mechanisms underlying these differences, it provides behavioral constraints on where specificity and generalization may arise within the olfactory system. Further neuroimaging and neurophysiological work will be needed to fully elucidate the underlying mechanisms.

One can suggest several straightforward extensions to the study that will make this study more influential. For example, generalization of odor detection threshold, generalization of perception (similar to androstenone), discriminating between two different odor concentrations. If these are also generalized across nostrils, then one can suggest a hypothesis as to why enantiomers don't. Other results will raise some other exciting hypotheses.

We thank the reviewer for the insightful suggestions. As noted on p.4 of the original manuscript, the primary aim of the present study was to test whether olfactory learning can generalize to odors distinct from those used in training—an issue that, to our knowledge, has not been experimentally addressed before. In examining cross-odor generalization, we also assessed whether learning transferred across nostrils.

Several studies (Bremner et al., 2003; Mainland et al., 2002; Wang et al., 1993; Yee and Wysocki, 2001) have examined potential mechanisms underlying olfactory sensitization to androstenone and other odorants (e.g., amyl acetate). Lesion and neurophysiological studies in mice suggest peripheral involvement in exposure-induced reductions in detection thresholds (Wang et al., 1993; Yee and Wysocki, 2001). In humans, the classification of individuals as androstenone detectors or non-detectors typically relies on criteria that favor type II error (Bremner et al., 2003). It is therefore conceivable that the shift from non-detection to detection observed in earlier odor-exposure studies partly reflects hyposmic individuals becoming more aware of the target odor (androstenone), which would implicate higher-order levels of the olfactory hierarchy. Whether this is the case, however, lies beyond the scope of the current study.

We agree with the reviewer that the mixture discrimination task could, in principle, reduce to discriminating relative concentrations of the constituent components. Following the reviewer’s suggestion, we conducted a new experiment that directly examined concentration discrimination learning using the same procedures as in Experiments 1 and 2. In contrast to mixture discrimination learning, however, training with different concentrations of a single compound produced neither cross-nostril transfer nor generalization to novel odorants, and the training effect was short-lived, dissipating within two weeks. These findings led us to conclude that concentration discrimination learning cannot account for the generalization and persistence observed in mixture discrimination; rather, the two forms of olfactory learning appear to be supported by distinct mechanisms. This new experiment has been incorporated as Experiment 3 and is detailed on p.12-15 of the revised manuscript.

We have updated the Discussion (p.16-18 of the revised manuscript) to include revised analyses and interpretations of differences in specificity, transfer, and persistence across mixture, enantiomer, and concentration discrimination learning. Please also refer to our response to a related point raised by Reviewer 2 (Point 3) below.

Reviewer #2 (Recommendations for the authors):The manuscript from Chang et al. taps on an important issue in olfactory perceptual plasticity, named the generalization of perceptual learning effect by training using odors. They employed a discrimination training/learning task with either binary odor mixture or odor enantiomers, and tested for post-training effect at several time intervals. Their results showed contrasting patterns of specificity (enantiomers) and transfer (odor mixtures), and the learning effect persisted at 2 weeks post-training. They demonstrated that the effect was independent of task difficulty, olfactory adaptation and gender.Overall this was a well-controlled study and shows novel results. The strength of the study includes the consideration of odor structure and perceptual (dis)similarity and the control training condition.I suggest that the authors discuss their work in relation to works on central representation of certain features of odors (valence, identity).

We appreciate the reviewer’s suggestion and have expanded the Discussion (p.16-18 of the revised manuscript) to relate our findings to prior work on central representations of odor valence, identity and other perceptual properties. Specifically, we now include the following passage:

“The nostril- and structure-specificity of enantiomer and concentration discrimination learning likely reflects modifications in or upstream of the olfactory bulb (Feng and Zhou, 2019). Conversely, the inter-nostril transfer of mixture discrimination learning suggests that learning occurs at a stage where binaral inputs converge, i.e., beyond the primary olfactory cortex. The generalization of mixture discrimination learning to unrelated mixtures further indicates the involvement of abstract representations of discrimination boundaries for inputs varying in odor quality. A plausible locus of plasticity underlying such generalization is the orbitofrontal cortex, the principal neocortical target of primary olfactory areas (Gottfried, 2010). This region is involved in the perceptual coding of odor identity (Li et al., 2006), valence (Anderson et al., 2003), and other properties, and plays a key role in olfactory discrimination (Li et al., 2006; Zatorre and Jones-Gotman, 1991). In particular, it has been proposed to integrate olfactory perceptual evidence toward a decision criterion in mixture discrimination (Bowman et al., 2012). Prolonged training with odor mixtures may refine the representation of perceptual evidence (e.g., relative differences in configural odor notes) in the orbitofrontal cortex and modify the discrimination boundary (Qamar et al., 2013). Applying this optimized boundary could yield generalized improvements in mixture discrimination beyond the trained nostril and training pair.”

References

Anderson, A. K., Christoff, K., Stappen, I., Panitz, D., Ghahremani, D. G., Glover, G., Gabrieli, J. D., and Sobel, N. (2003). Dissociated neural representations of intensity and valence in human olfaction Nat Neurosci, 6(2), 196-202.

Bowman, N. E., Kording, K. P., and Gottfried, J. A. (2012). Temporal integration of olfactory perceptual evidence in human orbitofrontal cortex Neuron, 75(5), 916-927.

Bremner, E. A., Mainland, J. D., Khan, R. M., and Sobel, N. (2003). The prevalence of androstenone anosmia Chem Senses, 28(5), 423-432.

Feng, G., and Zhou, W. (2019). Nostril-specific and structure-based olfactory learning of chiral discrimination in human adults *eLife*, 8.

Gottfried, J. A. (2010). Central mechanisms of odour object perception Nat Rev Neurosci, 11(9), 628-641.

Li, W., Luxenberg, E., Parrish, T., and Gottfried, J. A. (2006). Learning to smell the roses: Experience-dependent neural plasticity in human piriform and orbitofrontal cortices Neuron, 52(6), 1097-1108.

Mainland, J. D., Bremner, E. A., Young, N., Johnson, B. N., Khan, R. M., Bensafi, M., and Sobel, N. (2002). Olfactory plasticity: One nostril knows what the other learns Nature, 419(6909), 802.

Qamar, A. T., Cotton, R. J., George, R. G., Beck, J. M., Prezhdo, E., Laudano, A., Tolias, A. S., and Ma, W. J. (2013). Trial-to-trial, uncertainty-based adjustment of decision boundaries in visual categorization Proc Natl Acad Sci U S A, 110(50), 20332-20337.

Wang, H. W., Wysocki, C. J., and Gold, G. H. (1993). Induction of olfactory receptor sensitivity in mice Science, 260(5110), 998-1000.

Yee, K. K., and Wysocki, C. J. (2001). Odorant exposure increases olfactory sensitivity: Olfactory epithelium is implicated Physiol Behav, 72(5), 705-711.

Zatorre, R. J., and Jones-Gotman, M. (1991). Human olfactory discrimination after unilateral frontal or temporal lobectomy Brain, 114 (Pt 1A), 71-84.